# Effect of Conventional Preservatives and Essential Oils on the Survival and Growth of *Escherichia coli* in Vegetable Sauces: A Comparative Study

**DOI:** 10.3390/foods12152832

**Published:** 2023-07-26

**Authors:** Kateřina Hanková, Petra Lupoměská, Pavel Nový, Daniel Všetečka, Pavel Klouček, Lenka Kouřimská, Miroslava Hlebová, Matěj Božik

**Affiliations:** 1Department of Food Science, Czech University of Life Sciences Prague, Kamýcká 129, 165 00 Prague, Czech Republicp.lupomeska@seznam.cz (P.L.); novy@af.czu.cz (P.N.); kloucek@af.czu.cz (P.K.); 2Palíto Family s.r.o., Kamýcká 1281, 165 00 Prague, Czech Republic; chilliomackypalito@gmail.com; 3Department of Microbiology Nutrition and Dietetics, Czech University of Life Sciences Prague, Kamýcká 129, 165 00 Prague, Czech Republic; kourimska@af.czu.cz; 4Department of Biology, University of Ss. Cyril and Methodius in Trnava, Nám. J. Herdu 2, 917 01 Trnava, Slovakia; miroslava.hlebova@ucm.sk

**Keywords:** *E. coli*, vegetable sauce, essential oil, preservation

## Abstract

Essential oils have gained attention as natural alternatives to chemical preservatives in food preservation. However, more information is needed regarding consumer acceptance of essential oils in actual food products. This study aimed to compare the effects of conventional preservatives, heat treatment, and essential oils derived from thyme, oregano, and lemongrass on the survival and growth of pathogenic *Escherichia coli* in vegetable sauces. The results demonstrated a gradual decrease in pathogen numbers over time, even in untreated samples. On the fifth day of storage, heat treatment, sodium chloride, and acidification using citric acid (pH 3.2) exhibited reductions of 4.4 to 5.3 log CFU/g compared to the untreated control. Among the essential oils tested, lemongrass essential oil at a concentration of 512 mg/kg demonstrated the most remarkable effectiveness, resulting in a reduction of 1.9 log CFU/g compared to the control. Fifteen days after treatment, the control samples exhibited a contamination rate of 6.2 log CFU/g, while *E. coli* numbers in treated samples with heat, sodium chloride, citric acid (pH 3.2), and lemongrass essential oil (512 mg/kg) were below the detection limits. Additionally, sensory evaluation was conducted to assess the acceptability of the treated samples. The findings provide valuable insights into the potential utilisation of essential oils as natural preservatives in vegetable sauces and their impact on consumer acceptance.

## 1. Introduction

Essential oils (EOs) are considered an economical, eco-friendly, and natural alternative to chemical preservatives commonly used in food preservation [1]. Increasingly, consumers are demanding fresher, minimally processed foods without chemical preservatives. However, maintaining food safety and microbial quality in food production presents significant challenges. Although numerous studies have been focused on the use of EOs in the food industry [2], there still needs to be more information on the acceptance of essential oils by consumers in real products [2,3]. Effective concentrations of EOs often exceed the threshold of consumer acceptability [4]. In recent years, outbreaks of *Salmonella* sp. and *Escherichia coli* originating from minimally processed or fresh foods have been reported [5,6,7]. Salsa has frequently been identified as a vehicle for foodborne pathogens in the USA and South America, where it is widely consumed. *E. coli* is among the pathogens commonly associated with salsa contamination [8]. In the European Union, an estimated 44 outbreaks of microbial diseases associated with the consumption of fresh produce were reported between 1999 and 2019, with 64% of cases linked to the consumption of contaminated vegetables and salads [9]. The microorganism not only exhibits resilience to the low pH levels typically presented in vegetable sauces but also demonstrates prolonged persistence when contaminated products are stored at refrigerated temperatures within such an environment. This enables its survival in the low-pH conditions characteristic of vegetable sauces [10,11,12].

Sauces such as salsa and guacamole are prepared using ingredients like tomatoes, coriander, chillies, and onions, previously associated with the transmission of foodborne pathogens [8]. Contaminated salsa can provide a conducive environment for the growth of pathogenic microorganisms, especially when stored at room temperature for extended periods. Common pathogenic bacteria found in vegetable and other sauces include *E. coli*, *Salmonella* sp., and *Listeria monocytogenes*, with strains that exhibit tolerance to acid conditions, enabling survival in these products [9]. For instance, *E. coli* O15:H7 can withstand pH as low as 4 [13]. Moreover, according to studies, it can persist for weeks in acidic products when stored at refrigerated temperatures, as lower temperatures enhance its survival duration [14]. Several studies have been published focusing on the synergistic effects of microwave heating and EOs in chilli sauce [15], the protective properties of EO in marinades for chicken [16] or beef meat [17], the sensory attributes of habanero chilli pastes with natural preservatives and thermal processing [18], the antifungal effect of coriander [19] and cinnamon [20] EOs in tomato sauce, and the use of cinnamaldehyde in combination with acetic acid to reduce *E. coli* and *Salmonella* sp. on spinach leaves [21]. Most published studies thus far present EOs as effective and potential substitutes for conventional preservation techniques. However, comparative analyses with conventional preservation methods such as temperature and organic acids and their salts are frequently lacking. Therefore, in this study, we aimed to compare the effects of conventional preservatives, heat treatment, and various types of EOs on the survival and growth of *E. coli* in vegetable sauces.

## 2. Materials and Methods

### 2.1. Microbial Strain

The bacterial strain used for inoculation was *Escherichia coli* (ATCC 25922). The liquid culture medium, TSB (Tryptone soya broth Oxoid, Basingstoke, UK), was used to prepare the bacterial inoculum with 1% glucose addition (Sigma-Aldrich, St. Louis, MO, USA). The glucose fermentation in the medium resulted in a decreased pH, thereby increasing the bacteria’s resistance towards acidic environments, as reported by a previous study [22]. The bacterial culture was incubated at 37 °C for 16 h.

### 2.2. Preservatives and Essential Oils

Various conventional preservatives that are commonly used in the food industry, including sodium chloride (Lach-Ner, Neratovice, CZ), sodium benzoate (Sigma-Aldrich, St. Louis, MO, USA), sorbic acid (Sigma-Aldrich, St. Louis, MO, USA), sucrose (Lach-Ner, Neratovice, CZ), and citric acid (Lach-Ner, Neratovice, CZ), were employed in this experiment. In addition, essential oils (EOs) of thyme (*Thymus vulgaris*, Sigma-Aldrich, St. Louis, MO, USA), oregano (*Origanum vulgare*, Biomedica, Prague, CZ), and lemongrass (*Cymbopogon citratus*, Biomedica, Prague, CZ) and freshly squeezed lime juice were also used as additives due to their well-established antibacterial activity. The compositions of EOs were previously analysed using GC/MS and reported [23,24]. The GC-MS analyses were performed using an Agilent 7890A GC coupled with an Agilent MSD5975C MS detector (Agilent Technologies, Palo Alto, CA, USA). Thymol (44%) and p-cymene (18%) were found to be the major components of thyme oil, while carvacrol (70%) was the main constituent of oregano essential oil. The major components of lemongrass oil were geranial (40%) and neral (32%).

### 2.3. Salsa Preparation and Inoculation

The salsa used in this study was prepared from fresh vegetables purchased from a local grocery market (Kaufland Czech Republic v.o.s., Prague, CZ). Each ingredient was accurately weighed using a digital scale (Kern KB 2400-2N, Großmaischeid, DE), thoroughly washed, dried, and then chopped to the desired consistency using a kitchen blender (Bosh MCM3200W, Gerlingen-Schillerhöhe, DE). The resulting sauce had the following weight ratio of ingredients: 65% red tomatoes, 15% kitchen onions, 10% red peppers, 5% jalapeño peppers, and 5% rawitt peppers. Then, 18 mL of overnight grown *E. coli* culture with an approximate concentration of 10^9^ CFU/mL [25] was added to 900 mL of prepared salsa. The mixture was then thoroughly mixed using a laboratory electric mixer (Steinberg Systems SBS-ER-3000, Berlin, DE) [26].

Subsequently, individual 20 mL samples were taken from the inoculated salsa (Table 1). The following preservatives: sodium chloride, sodium benzoate, sorbic acid, sucrose, citric acid, essential oils of thyme, oregano, and lemongrass, and freshly squeezed lime juice were added to the prepared samples. To do this, each preservative was added to a measured amount of salsa in a beaker, and the mixture was thoroughly mixed using a laboratory electric mixer. The prepared sauce samples were carefully transferred into tightly closed samplers and stored at 4 °C for subsequent analysis. The presence of *E. coli* in the samples was analysed on days 1, 3, 5, 7, 10, and 15 after treatment. Heat treatment was used as a control. The sample was heated in a water bath until the temperature inside the sample reached 90 °C in all parts of the sauce. All samples were prepared in triplicate.

### 2.4. Determination of Microbial Contamination

The isolation and identification of *E. coli* were conducted using the ISO 16649 standard microbiological method on TBX medium and easySpiral—Automatic plater (Interscience, Saint Nom la Brétèche, FR). Individual samples of inoculated salsa were aseptically mixed with a sterile metal spoon within a laminar box and weighed into sealed plastic tubes using a digital balance. Phosphate-buffered saline was added to the weighed samples to achieve the desired dilution of 1:10. The diluted samples were briefly shaken on a vortex shaker at 2500 rpm for five seconds and transferred into sterile microtubes. The contents of the microtubes were then plated onto Petri dishes containing prepared TBX medium (Oxoid, Basingstoke, UK) using Automatic Plater. Subsequently, the Petri dishes were dried and incubated at 37 °C for 24 h. On the second day, colony enumeration was performed following the instructions provided in the documentation for the spiral inoculator [27]. Each plate was assessed by counting colonies in opposite quadrants, with two evaluations conducted for each plate. Microbiological analyses were carried out to evaluate the survival and growth of pathogenic *E. coli* in the inoculated salsa treated with different methods and subsequently stored at 4 °C. The analyses were performed on days 1, 3, 5, 7, 10, and 15 following the treatment. The objective of this study was to determine whether the selected treatments resulted in a significant reduction in the microbiological contamination of the vegetable sauce.

### 2.5. Sensory Analysis

Based on the assessment of the microbiological outcomes, the most promising treatments, along with untreated control, were selected for a sensory evaluation (refer to Table 2). The following samples were presented to the assessors: a heat-treated sample, samples supplemented with sodium benzoate, sodium chloride, citric acid, thyme, and lemongrass EOs (at a concentration of 512 μL/L), and one control sample without any treatment. The specific variations examined are detailed in Table 2. On the day of the sensory analysis, 1000 mL of fresh salsa was prepared 3 h prior to the evaluation according to the procedure outlined in Section 2.3. The salsa preparation and inoculation. No bacteria were added to the samples for tasting. Samples were divided into eight 125 mL beakers and then treated by the selected methods.

Sensory evaluation was conducted immediately following the completion of sample preparation in the sensory laboratory of the Czech University of Life Sciences Prague, ensuring compliance with the requirements specified in ISO standard ISO 8589:2007. The various sauce variants were served in 5 cm diameter glass bowls labelled with randomly assigned four-digit codes. Each dish was served with 100 mL of sample. The bowls containing all the sauce variants were then arranged randomly on plastic trays. In addition to the set of samples accompanied by plastic spoons, the evaluators were provided with a plate containing 25 g of corn tortilla chips (Snack Day, BE) and beakers of water and 30% ethanol, which served as flavour neutralisers. To ensure the reliability of the sensory panel, individuals with a negative predisposition towards spicy foods containing chilli peppers were excluded from participation. The sensory evaluation is employed as a sensory profile method based on ISO 13299:2016. Four descriptors were evaluated for each sauce variant using unstructured graphical scales of 100 mm length: pleasantness of aroma (0 = disgusting, 10 = very pleasant), pleasantness of taste (0 = disgusting, 10 = very pleasant), intensity of spicy flavour pungency (0 = unnoticeable, 10 = very strong), and overall rating of the sample (0 = completely unacceptable, 10 = excellent). Additionally, a hedonic ranking test based on ISO 8587:2006 was performed, in which the panellists ranked the submitted set of samples in order of increasing pleasantness/acceptability, from the least acceptable to the most acceptable sample. A total of 12 trained evaluators, encompassing both men and women from various age groups, participated in the sensory evaluation.

### 2.6. Statistical Evaluation

The data from the experimental phase were processed and statistically analysed using MS Excel and Statistica 12. To facilitate the analysis, the results of microbiological measurements were initially transformed from CFU/g to log CFU/g. After testing the assumptions of normality of the data and homogeneity of variances, a one-way analysis of variance (ANOVA) followed. Scheffé’s method was chosen for the post hoc analysis. Furthermore, Friedman’s test was used to evaluate the ranking test. All statistical testing was performed at a significance level of α = 0.05.

## 3. Results

The microbiological analysis aimed to evaluate the survival and growth of pathogenic *E. coli* in inoculated salsa treated using various treatment methods and stored at 4 °C. The analysis was conducted at specific time intervals, namely days 1, 3, 5, 7, 10, and 15 following treatment. The experiment was divided into two parts, and separate evaluations were performed for each part. The obtained results were subjected to statistical analyses (Table 3).

Throughout the experiment, a gradual decrease in the number of pathogens was observed in all samples, including the untreated ones. On the first day, only samples treated with heat, 10% sodium chloride, 60% sucrose, and citric acid at a final pH of 3.2 exhibited detectable colonies on the nutrient medium. Among these, the heat-treated samples displayed the lowest pathogen count. By the third day after treatment, colonies were also detectable in samples treated with 30% sucrose. Between the third and fifth day, a significant reduction in *E. coli* counts occurred, enabling the determination of log CFU/g values for all samples. The sauce treated with heat, citric acid (pH 3.2), and sodium chloride exhibited the lowest counts.

Moreover, significantly lower pathogen levels were observed in salsa treated with both 30% and 60% sucrose, as well as thyme oil at a concentration of 64 µL/L, compared to the untreated samples. The pathogen count dropped below detectable levels in the heat-treated samples from the fifth to the seventh day. Similar to previous days, samples treated with citric acid (pH 3.2) and sodium chloride exhibited the lowest log CFU/g values. Significant reductions in bacterial counts compared to untreated salsa were also observed in samples treated with sucrose (both 30% and 60%) and sorbic acid. Between the seventh and tenth day, *E. coli* counts fell below detectable levels in the samples treated with sodium chloride. However, except for the sauce treated with thyme oil and thyme at a concentration of 34 µL/L, all treatments on day 10 showed significantly lower log CFU/g values than the untreated samples. On the last monitoring day, bacterial presence was not detected in samples treated with heat and sodium chloride or acidified with citric acid to reach pH 3.2. Among the other treatments, adding 60% and 30% sucrose resulted in a substantial reduction in pathogen contamination. In contrast, samples treated with 34 µL/L of thyme essential oil exhibited the least reduction, with the log CFU/g even higher than that of the untreated samples on day 15, although the difference was insignificant.

The sensory analysis aimed to assess the impact of different treatments on the sensory attributes of salsa. The sensory profile method was employed to evaluate the aroma pleasantness, pleasantness of taste, intensity of pungency, and overall acceptability of the samples. The outcomes of the evaluation, along with the post hoc tests, are presented in Table 4. The analysis of variance revealed significant differences in the pleasantness of aroma, pleasantness of taste, and overall rating of the samples.

In general, the sample treated with thyme essential oil at a concentration of 512 µL/L received the lowest average scores when assessing aroma pleasantness, taste pleasantness, and overall acceptability. Conversely, the sample treated with citric acid at a pH of 3.52 obtained the highest scores in terms of these three descriptors. Regarding evaluating the pungency intensity, the sauces treated with citric acid at pH 3.2 and lemongrass essential oil at 512 µL/L were perceived as the most intensely pungent on average. In contrast, the salsa sample with sodium chloride was rated as the least spicy. However, the differences in pungency intensity between the samples were not found to be statistically significant. During the hedonic ranking test, the evaluators ranked the series of submitted samples in order of increasing acceptability, from the least acceptable to the most acceptable sample, using Friedman’s test for evaluation. The calculated value of Friedman’s criterion (31.56) exceeded the critical value of Friedman’s criterion (13.73), and thus the overall difference between the samples was significant. For further comparisons between samples, it was determined that there was a significant difference when the absolute value of the differences in the sum of the rankings of the samples exceeded 23.52 (Table 5). Overall, the treatments with citric acid and sodium benzoate received the highest scores. In contrast, thyme essential oil received the lowest scores, showing significant distinctions from all other treatments except the sodium chloride treatment. Additionally, the treatment with lemongrass essential oil did not negatively impact the sensory parameters compared to the untreated control. 

## 4. Discussion

Vegetable sauces have gained significant recognition for their ability to preserve fruit and vegetables. The preservation process typically involves the addition of sugar or acid to the mixture and subsequently reducing water content through heating, which are fundamental components of the preservation process. Preservatives such as benzoic acid, sorbic acid, and their salts may be added to enhance durability and safety. A modern trend involves using EOs as substances that contribute to the extended shelf life of food products. Although the antimicrobial activity of EOs and their active compounds has been extensively investigated, their practical application in the market is not yet fully evident. The bioactivity of EOs is generally attributed to their phenolic compounds, which are soluble in the lipid layer of membranes and impact membrane fluidity.

The use of EOs as preservatives often requires their application in high concentrations to achieve effective preservation, which can lead to undesirable sensory changes [28]. During sensory evaluations, panellists have reported perceiving a sour taste and strong chemical or herbal aroma [4]. To enhance the antimicrobial effectiveness of EOs, researchers have explored combining them with physical methods such as ohmic heating [29] or microwave heating [15]. The consumer shift towards natural antimicrobials has contributed to a notable change in attitudes towards synthetic preservatives, leading to an increasing demand for natural alternatives. Our study conducted a comparative analysis by incorporating synthetic preservatives and heat treatment alongside natural preservatives. Sodium benzoate and sorbic acid were used at their maximum permitted levels [30,31], while sucrose was used at concentrations of 30% and 60% for evaluation purposes. The selection of sauces was based on market research, which revealed that sweet and sour chilli sauces typically contain sugar content ranging from 30 to 70 g per 100 g. Citric acid and its alternative, lime juice, were used at values below and above pH 4. The most effective preservation method was demonstrated to be heat treatment at 90 °C for 1 min. Although pathogen elimination was not achieved, the remaining population fell below detectable levels between days 5 and 7, the earliest compared to other samples. However, it is important to note that heat treatment has inherent limitations, such as the potential for the loss of nutritional and sensory quality, which contradicts the increasing consumer preference for fresher, higher-quality, and healthier food options [13]. On the fifth day following treatment, *E. coli* was detected in all samples. Heat treatment, sodium chloride, and acidification using citric acid (pH 3.2) exhibited a reduction of 2 log CFU/g, whereas the other treatments demonstrated a more substantial reduction of 6–7 log CFU/g. Among the EOs tested, lemongrass essential oil at a concentration of 512 mg/kg exhibited the highest efficacy, resulting in a detection of 5 log CFU/g, which represented a reduction of 1–2 log CFU compared to the other samples.

After fifteen days of treatment, the control samples exhibited 6 log CFU/g. In contrast, samples treated with heat, sodium chloride, citric acid (pH 3.2), and lemongrass essential oil at a concentration of 512 mg/kg exhibited *E. coli* counts below the detection limit [13]. It has been demonstrated in a previous study [32] that combining essential oil with sodium chloride exhibits bactericidal effects of carvacrol and thymol against *E. coli.*, leading to noteworthy enhancements in antimicrobial activity. In another study, nanoemulsions formulated with various concentrations of oregano were tested in situ for antifungal activity against *Zygosaccharomyces bailii* in different salad dressings. The reference samples maintained constant microbial counts at 5 log CFU/g. However, incorporating the oregano and clove nanoemulsions into salad dressings led to a reduction in fungal count compared to the reference sample, with reductions of 1 and 2 log CFU/g, respectively [33]. Moreover, the treatment of chilli sauce with microwave heating in combination with essential oil components (carvacrol, eugenol, carvone, and citral) resulted in reductions in *E. coli,* ranging from 1.6 to 4.5 log/mL [15]. Additionally, when chicken samples were marinated with 1% and 2% carvacrol or thymol, there was a decrease in *E. coli* O157:H7 numbers during storage by approximately 3.3 log CFU/g, in comparison to unmarinated samples [16]. In our specific case, treatment with oregano oil did not significantly reduce *E. coli* counts.

In the experiment, sorbic acid at a concentration of 715 mg/L and sodium benzoate at a concentration of 1000 mg/L were employed as chemical preservatives. Adding these preservatives resulted in a significant decrease in bacterial counts compared to untreated samples. However, it should be noted that the pathogen was not reduced to undetectable levels during the 15-day monitoring period. On the last day, the salsa treated with sorbic acid exhibited a count of 5.77 log CFU/g, while the sample with sodium benzoate had a count of 2.77 log CFU/g. Some studies suggest that sodium benzoate might be more efficient than potassium sorbate, the commonly used salt of sorbic acid, in inactivating pathogens like *E. coli* O157:H7 [34]. This efficacy of sodium benzoate has been previously demonstrated in studies on apple ciders and juices, where it led to a reduction in *E. coli* levels below detectable limits in a shorter timeframe compared to potassium sorbate [10]. These findings support the effectiveness of preservatives in reducing pathogen survival in acidic plant products, including salsa. However, when selecting preservatives, careful consideration should be given to their impact on sensory attributes and consumer preferences.

The antimicrobial activity of EOs, as well as the effectiveness of their active compounds, has been extensively investigated [35]. The bioactivity of EOs is generally attributed to phenolic compounds (phenols), which are soluble in the lipid layer of the membrane and alter membrane fluidity [36,37]. However, achieving a sufficient preservative effect requires the use of high concentrations, which usually cause organoleptic changes. In particular, sensory evaluations have reported a sour taste and intense chemical or herbal aroma [38]. Another approach to increase the antimicrobial effectiveness of EOs is to combine them with various physical methods such as ohmic heating [9] or microwave heating [39].

In addition to natural preservatives, synthetic preservatives and heating were included in the testing to enable comparison. Although heating under the selected parameters did not result in the complete elimination of the pathogen, the remaining population fell below detectable levels between days 5 and 7, earlier than the other samples. Despite its high availability, efficacy, and low cost, heating remains the predominant method of food preservation. However, its main drawbacks include the loss of nutritional and sensory quality of the product, which contradicts the increasing consumer interest in fresher, higher-quality, and healthier foods [9]. The addition of synthetic preservatives resulted in a significantly greater decrease in log CFU/g compared to untreated samples, but in neither case was the bacterium reduced below detectable levels during the 15-day monitoring period. On the last day, the salsa treated with sorbic acid exhibited a count of 5.77 log CFU/g, while the sample with sodium benzoate had a count of 2.77 log CFU/g. Although this difference is also related to the use of different concentrations, results from some studies have indicated that sodium benzoate demonstrates higher efficiency than potassium sorbate, a frequently used salt of sorbic acid, in inactivating pathogens such as *E. coli* O157:H7 [34]. Comparative assessments of the effects of benzoates and sorbates on *E. coli* survival in acidic plant products have been conducted in the past, particularly in apple ciders and juices. Zhao et al. (1993) [10] tested apple ciders (pH 3.6–4.0) inoculated with *E. coli* O157:H7 at 5 log CFU/mL, treated with 0.1% sodium benzoate or potassium sorbate and subsequently stored at 8 °C. The pathogen exhibited survival for a period of 10–31 days in untreated ciders. Potassium sorbate had a relatively minor effect, as *E. coli* dropped below detectable levels after 15–20 days.

In contrast, when treated with sodium benzoate, this timeframe was reduced to 2–10 days. The higher efficacy of sodium benzoate is also supported by the findings of Ceylan et al. [40]. In their research, apple juice (pH 3.75) was used, and inoculated samples were also treated with 0.1% sodium benzoate or potassium sorbate, then stored at 8 °C for 14 days. The population of *E. coli* O157:H7 decreased from the initial count of 5.2 log CFU/mL to 0.3 log CFU/mL when treated with sodium benzoate and 1.4 log CFU/mL when treated with potassium sorbate. The presence of the bacteria beyond 14 days, despite the high storage temperature and significantly lower pH of the product compared to the salsa employed (pH 4.58–4.6), demonstrates the notable ability of the bacteria to persist under experimental conditions, thus indicating the effectiveness of the preservatives.

## 5. Conclusions

The results demonstrated that heat treatment at 90 °C for 1 min proved to be the most effective method, leading to a significant reduction in *E. coli* counts and achieving levels below detectable limits by the fifth to seventh day of storage. However, it is worth noting that this treatment may compromise the nutritional and sensory quality of the product. Among the chemical preservatives tested, sorbic acid at a concentration of 715 mg/L and sodium benzoate at a concentration of 1000 mg/L exhibited effectiveness in reducing *E. coli* counts compared to untreated samples. Although the pathogen was not eliminated, the use of these preservatives resulted in a significant decrease in bacterial counts during the 15-day monitoring period.

Similarly, the application of EOs, specifically oregano, thyme, and lemongrass treatments at various concentrations, showed a gradual reduction in *E. coli* survival over time. Higher concentrations of these EOs exhibited a more pronounced effect on reducing pathogen counts. Nevertheless, careful consideration should be given to the sensory changes associated with the use of high concentrations of EOs and synthetic preservatives. Consumer preferences are currently shifting towards natural antimicrobials, and there is an increasing demand for products with minimal synthetic preservatives.

Overall, this study highlights the potential of various treatments, including heat treatment, chemical preservatives, and EOs, in reducing the survival of pathogenic bacteria in salsa. Although prior research [2,3,41,42] has portrayed EOs as highly efficient and comprehensive substitutes for conventional preservation methods, a common limitation is the absence of comparative analyses with traditional preservation techniques involving temperature, organic acids, and their salts. Therefore, further research and optimisation of these treatments are necessary to develop effective and consumer-friendly strategies for improving the microbiological safety of salsa and similar products.

## Figures and Tables

**Table 1 foods-12-02832-t001:** Variants of treatment methods and conditions applied in vegetable salsa.

Treatment	Value
without treatment (pH 4.58)	-
without treatment (pH 4.6)	-
heat treatment	90 °C 1 min
heat treatment	90 °C 1 min
citric acid (pH 32)	145 g/L
citric acid (pH 352)	86 g/L
lime juice (pH 365)	135 mL/L
lime juice (pH 383)	90 mL/L
lime juice (pH 414)	45 mL/L
sucrose	300 g/L
sucrose	600 g/L
sodium benzoate	1 g/L
sodium chloride	100 g/L
sorbic acid	715 mg/L
thyme essential oil	512 μL/L
thyme essential oil	256 μL/L
thyme essential oil	128 μL/L
thyme essential oil	64 μL/L
thyme essential oil	32 μL/L
lemongrass essential oil	512 μL/L
lemongrass essential oil	256 μL/L
lemongrass essential oil	128 μL/L
lemongrass essential oil	64 μL/L
lemongrass essential oil	32 μL/L
oregano essential oil	512 μL/L
oregano essential oil	256 μL/L
oregano essential oil	128 μL/L
oregano essential oil	64 μL/L
oregano essential oil	32 μL/L

**Table 2 foods-12-02832-t002:** Samples prepared for sensory evaluation.

Method of Treatment
without treatment
heat treatment 90 °C 1 min
sodium benzoate 1000 mg/L
sodium chloride 100 g/L
citric acid pH 3.2
citric acid pH 3.52
lemongrass essential oil 512 μL/L
thyme essential oil 512 μL/L

**Table 3 foods-12-02832-t003:** *E. coli* log CFU/g sauce on each day of measurement (mean ± SD).

*	Day 1	Day 3	Day 5	Day 7	Day 10	Day 15
without treatment (pH 4.58)	ND	ND	7.11 ± 0.05 ^a^	6.99 ± 0.08 ^a^	6.77 ± 0.1 ^a^	6.19 ± 0.05 ^a^
without treatment (pH 4.6)	ND	7.14 ± 0.05 ^a^	6.98 ± 0.03 ^ab^	6.83 ± 0.06 ^ab^	6.6 ± 0.05 ^ab^	6.12 ± 0.03 ^ab^
heat treatment	2.78 ± 0.03	2.08 ± 0	1.78 ± 0 ^e^	ND	ND	ND
heat treatment	3.54 ± 0.02	2.68 ± 0	1.78 ± 0 ^e^	ND	ND	ND
citric acid (pH 3.2)	5.73 ± 0.05 ^a^	3.28 ± 0.03	2.47 ± 0.1 ^c^	2.00 ± 0.03 ^c^	1.78 ± 0.2	ND
citric acid (pH 3.52)	ND	6.97 ± 0.03 ^b^	6.4 ± 0.09 ^d^	5.97 ± 0.03 ^d^	5.22 ± 0.04 ^c^	3.15 ± 0.04 ^c^
sodium benzoate	ND	6.97 ± 0.07 ^bdfghijk^	6.36 ± 0.04 ^dgk^	6.06 ± 0.04 ^defir^	5.32 ± 0.08 ^co^	2.77 ± 0.06
sorbic acid	ND	ND	6.99 ± 0.06 ^abhjmps^	6.65 ± 0.05 ^bjlmopq^	6.27 ± 0.06 ^dgikln^	5.77 ± 0.07 ^fij^
lime juice (pH 3.65)	ND	6.99 ± 0.07 ^bdf^	6.52 ± 0.03 ^dfgk^	6.27 ± 0.09 ^efi^	5.83 ± 0.03 ^eh^	4.54 ± 0.03 ^g^
lime juice (pH 3.83)	ND	7.08 ± 0.05 ^abfg^	6.63 ± 0.08 ^fgkl^	6.44 ± 0.05 ^eij^	6.11 ± 0.04 ^di^	5.22 ± 0.05
lime juice (pH 4.14)	ND	ND	7.02 ± 0.03 ^abhjm^	6.91 ± 0.08 ^abghk^	6.59 ± 0.05 ^bfj^	5.94 ± 0.04 ^efh^
sucrose 60%	5.76 ± 0.03 ^a^	5.19 ± 0.09	4.99 ± 0.04 ^i^	4.87 ± 0.08	4.72 ± 0.07	3.98 ± 0.03
sucrose 30%	ND	5.93 ± 0.03 ^e^	5.75 ± 0.04	5.27 ± 0.04	5.02 ± 0.03 ^c^	4.26 ± 0.03
sodium chloride	4.34 ± 0.03	3.59 ± 0.04	2.62 ± 0.04 ^c^	2.11 ± 0.07 ^c^	ND	ND
LG 32	ND	ND	7.09 ± 0.03 ^abh^	6.99 ± 0.05 ^abg^	6.54 ± 0.03 ^f^	5.95 ± 0.04 ^e^
LG 64	ND	ND	7.04 ± 0.05 ^abhj^	6.91 ± 0.07 ^abh^	6.45 ± 0.03 ^bg^	5.83 ± 0.03 ^ef^
LG 128	ND	6.75 ± 0.05 ^c^	6.53 ± 0.08 ^df^	6.27 ± 0.07 ^e^	6.18 ± 0.1 ^d^	5.49 ± 0.08 ^d^
LG 256	ND	6.87 ± 0.03 ^bcd^	6.49 ± 0.03 ^dfg^	6.03 ± 0.07 ^def^	5.82 ± 0.08 ^e^	4.91 ± 0.07
LG 512	ND	5.82 ± 0.04 ^e^	5.14 ± 0.05 ^i^	3.68 ± 0.03	ND	ND
O 32	ND	ND	7.06 ± 0.04 ^abhjp^	6.93 ± 0.05 ^abghlmn^	6.67 ± 0.07 ^abfjm^	6.2 ± 0.04 ^bk^
O 64	ND	ND	7.01 ± 0.04 ^abhjpr^	6.9 ± 0.07 ^abgklmnp^	6.61 ± 0.04 ^abfgjlm^	5.99 ± 0.03 ^ehijl^
O 128	ND	7.02 ± 0.07 ^bfgh^	6.76 ± 0.06 ^n^	6.73 ± 0.07 ^bhkl^	6.37 ± 0.04 ^dfgk^	5.86 ± 0.04 ^efhi^
O 256	ND	6.99 ± 0.08 ^bfghi^	6.78 ± 0.03 ^lno^	6.72 ± 0.08 ^bhklm^	u6.41 ± 0.03 ^bfgjkl^	5.89 ± 0.03 ^efhij^
O512	ND	6.85 ± 0.05 ^bcdfij^	6.74 ± 0.09 ^lnoq^	6.66 ± 0.09 ^bhijklo^	5.96 ± 0.03 ^ehi^	4.56 ± 0.07 ^g^
TH 32	ND	ND	7.03 ± 0.05 ^abhjmprstv^	6.89 ± 0.06 ^abghklmnopqs^	6.79 ± 0.08 ^abjm^	6.23 ± 0.06 ^abk^
TH 64	ND	ND	6.97 ± 0.04 ^abhjmprstv^	6.84 ± 0.06 ^bghklmnopq^	6.38 ± 0.07 ^dfgklnpq^	5.98 ± 0.04 ^ehijlm^
TH 128	ND	7.15 ± 0.05 ^ahl^	6.94 ± 0.05 ^bhjmoprst^	6.83 ± 0.08 ^abgklmnopqs^	6.22 ± 0.03 ^diklnp^	5.97 ± 0.03 ^ehijlm^
TH 256	ND	7.07 ± 0.03 ^abfhikl^	6.85 ± 0.06 ^bnoqrst^	6.71 ± 0.09 ^bklmnopqs^	6.27 ± 0.09 ^dgiklnpq^	5.57 ± 0.03 ^d^
TH 512	ND	6.85 ± 0.05 ^bcdfik^	6.59 ± 0.07 ^fgklq^	6.25 ± 0.07 ^efijr^	5.33 ± 0.04 ^o^	3.25 ± 0.03 ^c^

* LG = lemongrass essential oil; TH = thyme essential oil. Means ± standard deviation from three replications. Values followed by the same letters within the same column are not significantly different (*p* > 0.05).

**Table 4 foods-12-02832-t004:** Results of the evaluation of selected descriptors depending on the type of sample treatment.

Treatment	Descriptors (Mean ± SD) *			
	Pleasantness of Aroma (%)	Pleasantness of Taste (%)	Intensity of Pungency (%)	Overall Rating (%)
without treatment	64.29 ± 14.0 ^ab^	51.58 ± 14.4 ^ab^	41.92 ± 22.8 ^a^	49.54 ± 17.0 ^abc^
heat treatment	45.50 ± 18.7 ^ab^	49.96 ± 19.9 ^abc^	37.83 ± 22.4 ^a^	49.21 ± 19.8 ^abc^
sodium chloride	69.42 ± 19.7 ^ab^	28.58 ± 22.4 ^ac^	34.29 ± 19.5 ^a^	30.42 ± 20.5 ^bc^
TH 512 µL/L	41.42 ± 19.0 ^a^	21.46 ± 11.2 ^c^	43.54 ± 19.1 ^a^	23.79 ± 11.0 ^b^
LG 512 µL/L	53.58 ± 25.2 ^ab^	40.46 ± 22.9 ^abc^	53.00 ± 15.4 ^a^	45.25 ± 20.5 ^abc^
citric acid pH 3.2	61.42 ± 14.7 ^ab^	55.21 ± 20.7 ^ab^	53.29 ± 19.1 ^a^	58.96 ± 19.7 ^a^
citric acid pH 3.52	70.42 ± 11.3 ^b^	62.25 ± 18.0 ^ab^	50.83 ± 19.3 ^a^	68.04 ± 18.0 ^a^
sodium benzoate	64.46 ± 17.1 ^ab^	55.00 ± 19.3 ^ab^	44.08 ± 22.6 ^a^	58.17 ± 17.1 ^ac^

* LG = lemongrass essential oil; TH = thyme essential oil. Means ± standard deviation from three replications. Values followed by the same letters within the same column are not significantly different (*p* > 0.05).

**Table 5 foods-12-02832-t005:** Results of the comparison of individual samples according to Friedman’s test. Ranking totals and their relative differences.

			Without Treatment	Heat Treatment	Sodium Chloride	TH 512 µL/L	LG 512 µL/L	Citric Acid pH 3.2	Citric Acid pH 3.52	Sodium Benzoate
			**54**	**54**	**39**	**22**	**48**	**67**	**77**	**71**

without treatment	**54**		0	15	32	6	−13	−23	−17	−23
heat treatment	**54**		0	15	32	6	−13	−23	−17	−17
sodium chloride	**39**		−15	0	17	−9	−28	−38	−32	−17
TH 512 µL/L	**22**		−32	−17	0	−26	−45	−55	−49	−32
LG 512 µL/L	**48**		−6	9	−26	0	−19	−29	−23	−49
citric acid pH 32	**67**		13	28	45	19	0	−10	−4	−23
citric acid pH 352	**77**		23	38	55	29	10	0	6	−4
sodium benzoate	**71**		17	32	49	23	4	−6	0	6

LG = lemongrass essential oil; TH = thyme essential oil. Significant differences between samples are highlighted in red.

## Data Availability

Data are contained within the article.

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
