# Peer review of "Effect of Conventional Preservatives and Essential Oils on the Survival and Growth of Escherichia coli in Vegetable Sauces: A Comparative Study"

_foods, 2023, doi:10.3390/foods12152832_

Round 1
Reviewer 1 Report
The research is timely and contributes to the development of the field. I cannot evaluate the statistics correctly because I am not involved in this field, but I think that the statistical methods used are appropriate, given the article I have read.
- in the heading of Table 3, the name of the E. coli should be put in italics;
- under Table 3, there should be a legend with the abbreviations used in the first column;
- references should be edited between the text and at the end in the references section;
- in Table 4, the capitalised sd (SD) should be corrected;
- the Table 1 heading layout line needs to be corrected;
- the numbering of the tables needs to be edited, as Table 1 is at the end of the text;
- other current studies should be added to the debate, comparing the results and linking the conclusions to the benefit of the user;
- in conclusion, a broader perspective on the applicability of this research to the target population should be added.
Author Response
All comments are answered in the document "Author's Reply to the Review Report".

Reviewer 2 Report
The authors evaluated the effect of conventional preservatives, heat treatment, and essential oils on the survival and growth of Escherichia coli in salsa.
Major comments
The authors need to improve the introduction section.
Method: the method used were not clearly described. 2.5 Sensory analysis. The first paragraph is not clear. The authors failed to provide a clear description of how the samples for sensory analysis were prepared. Is there an ethical approval for this study?
Conclusion: The conclusion needs to be improved. The authors evaluated consumers acceptability of the essential oil treated product but did not highlight this in the conclusion. Based on the findings (microbiological and sensory analysis), can essential oil be a potential preservative for this product.
What is the take home message? We already know that these treatments will have effects on microbial growth and survival. What new is the study contributing.
Comments
Line 83, correct error
Line 91, state the time intervals
Line 101, what previous analysis? Reference
Line 107, include reference
Line 115-117, a 8 samples? the description from line 115 - 117 is not clear. It should be explanatory even without the Table. Correct accordingly
Line 118, Table 8? There is no Table 8 in the text
Line 120, described in the 0 2.3. Correct accordingly
Line 120, the salsa… of course. And what? The sentence is incomplete. What additional information is the sentence passing to the reader?
Line 157-158, ‘’..with separate evaluations performed for two parts of the experiment.’’ This was not captured in the methodology section.
Line 191-192, correct the error in the sentence
Line 201, which essential oil do you refer to as fragrance essential oil. Specific for clarity
Line 203, least hot? Do you mean spicy?
Line 212, do you mean Table 5? Correct accordingly.
Line 212, separate the title of the Table from the text
Line 218, ‘’…achieving effective preservation often requires…’’ confusing. Do you mean preservation of essential of oil or using essential oil as a preservative? Correct sentence
Line 222, ‘’… has been explored.’’ By whom? add reference
Line 236, ‘’Some studies suggest…’’ cite the studies at the end of the sentence
Line 239, ‘’… previous studies on apple ciders and juices…’’ cite the study.
Line 242, correct sentence
Line 254-256, correct sentence
Line 282, correct citation style
Line 284-289, correct sentence
Table 3, 4 and 5, define the abbreviations used in the Table as a footnote.
English language editing is needed. A native English speaker should read through the manuscript.
Author Response

(The authors gave the same response as above.)

Reviewer 3 Report
Regard the attachment.

Author Response

(The authors gave the same response as above.)

Reviewer 4 Report
The manuscript entitled Effect of Conventional Preservatives, Heat Treatment, and Essential Oils on the Survival and Growth of Escherichia coli in Vegetable Sauces" compared various treatments for the preservation of vegetable sauces.
Abstract needs extensive revision, by including the comparison of results supported by numerical values such word decrease in growth does not describe the extend of inhibition. Similarly, findings provide valuable insights?? please highlight. Abstract should describe what methods were undertaken as well.
Introduction: Background information is not enough, authors should undertake types of essential oil and major components with antibacterial effects, moreover, compare the literature of different treatments undertaken (Preferable last 5 year literature).
Line 83 reference not found??
What was the reason of choosing specific concentration oof each preservative????
Various methods have no reference to validate the methodology undertaken.
How, E. coli was inoculated in samples??
Line 117: Greatest reduction of E. coli?? please rephrase and mention Log CFU reduction.
Sensory and microbial analysis, please provide valid reference.
All scientific names should be in italics (Title table 3).
Include statistical interpretation in table 3.
Chemical composition of essential oils should be compared to justify the preservation potential.
The treatments with EOs resulted in above 5 CFU/g during storage then how to claim the preservative potential.
The experimental design is very poor and manuscript lacks in novelty.
Language expression of manuscript is fine.
Author Response

(The authors gave the same response as above.)

Round 2
Reviewer 2 Report
The authors have relatively improved the manuscript. However, they need to read thoroughly to correct language and grammatical errors in the manuscript and improve the coherency especially paragraphs where new texts were inserted. They should delete repeated sentences.
Comments
Correct the title. The authors should delete Heat Treatment from the title because heat treatment is one of the conventional methods used in the study.
Line 21, correct “… CFU/g the …” to “…CFU/g on the …”
Line 273, change “full-fledged” to “potential”
Line 744 - 746, correct and restructure the sentence
Line 748, change “promote” to “prolong”
Line 755 - 756, correct the sentence. Delete “EOs. In process of sensory evaluations,”
Line 760 – 762, correct the sentence
Line 765, delete “content”
Line 767 – 768, correct sentence
Line 780 – 783, correct sentence
Line 826, 1% a 2%?
Line 830 – 831, correct sentence
Line 844, “Nowadays”? This is a known fact. Delete “nowadays”.
Line 847 – 848, “…which usually case organoleptic changes.”? Correct sentence
Line 851 – 852, repetition
Line 856, “…felt…? Correct accordingly.
Line 86 – 862, repetition
Line 927, “…the results of? Include the authors’ names . Ceylan et al. [43]
Line 929 – 935, correct the sentences
Line 939, “…hat…? Correct accordingly.
Line 951, eat treatment? Correct
The manuscript needs extensive editing of English language
Author Response
the entire manuscript has undergone a thorough grammatical and stylistic check

Reviewer 4 Report
The manuscript has been significantly revised as advised by the reviewer, however the revised version does not include tables.
Author Response
we apologize for the lack of tables. The tables were sent as a separate file directly to the editor.
